# The Power of Images and the Logics of Discovery in Psychiatric Care

**DOI:** 10.3390/brainsci13010013

**Published:** 2022-12-21

**Authors:** Giovanni Stanghellini

**Affiliations:** 1Department of Health Sciences, University of Florence,50139 Florence, Italy; giostan@libero.it; Tel.: +39-347-379-0707; 2Centro de Estudios de Fenomenologia y Psiquiatrías, Diego Portales University, Santiago 8370068, Chile

**Keywords:** causality, culture, diagnosis, images, emergentism, narratives, Warburg

## Abstract

This paper, aligned with contemporary thinking in terms of patient-centered care and co-creation of patient care, highlights the limitations of the reductionist approaches to psychiatry, offering an alternative, “emergent” perspective and approach. Assuming that psychopathological phenomena are essentially relational, what kind of epistemological framework and ‘logic of discovery’ should be adopted? I review two standard methods I call ‘*ticking boxes*’ and ‘*drafting arrows*’. Within the *ticking boxes* framework, the clinician’s main goal is to discover whether a patient showing psychopathological phenomena meets pre-given diagnostic criteria. The process of discovery can be compared to two people assembling a puzzle where the patient has the pieces and the interviewer has the image of the completed design. *Drafting arrows* consists in constructing pathogenetic diagrams that display linear causative relationships between variables connected by an arrow to other nodes. These explanatory narratives include psychodynamic (motivational) and biological (causal) diagrams. I argue for a third approach called ‘linking dots’, a method of discovery based on the emergent properties of psychopathological phenomena. I build on and develop the approach to images and discovery devised by art historian Aby Warburg in his atlas of images *Bilderatlas Mnemosyne*. The visual constellations created by Warburg in the panels of the *Bilderatlas* can be understood as a method to reveal the layers of memory and the web of relationships manifested in them, inviting the viewer to participate in the production of meanings, forging ever new connections between the images. It is the viewer’s acts of perception that draw relationships between singularities. I suggest that this method is of enormous significance in the context of today’s socio-cultural transformation processes and related forms of psychopathological conditions, which can no longer be comprehended using the categories of existing knowledge systems.

## 1. *Ticking Boxes* and *Drafting Arrows*: Two Standard Ways to Knowledge in Psychopathology and Their Limitations

*“Augustine, Augustine, quid quereris? Putasne brevi immettere vasculo mare totum?” (“Augustine, Augustine what do you seek? Do you think perhaps you can put the whole sea in your ship?”)*.[1]

*“The same song was repeated to me elsewhere; no one wanted to admit that science and poetry could be combined. It was forgotten that science came out of poetry and it was not considered that by changing the times these two could amicably find themselves with mutual advantage on a higher level”*.[2]

There are two basic methods of discovery at play in psychopathology and related research. I will call the first one *ticking boxes*; the second, *drafting arrows*.

In *ticking boxes*, the main epistemological tenet is that psychopathological disorders manifest themselves in characteristic sets of signs, symptoms and behaviors that are, in principle, accessible to question-and-answer techniques (see Table 1). The clinician’s main goal is to discover whether a patient showing psychopathological phenomena meets pre-given diagnostic criteria. From this perspective, discovery is a kind of inquiry which should conform to the technical–rational paradigm of natural sciences in which psychiatry, as a branch of bio-medicine, is positioned. Interviewing is conceived as a stimulus–response pattern of questions formulated in such a way as to reduce information variance [3] and elicit only “relevant” answers [4]. A special emphasis is given to inter-rater reliability, i.e., reducing disagreement between different clinicians. The aim is neither an in-depth understanding of the patient’s personal experiences, nor to reveal previously unknown features of the patient’s condition, but to assess those phenomena that are deemed important as diagnostic indexes a priori, leading to the classification of the patient’s complaints and dysfunctions according to pre-defined diagnostic categories.

Diagnostic criteria are often established in a “politically driven manner” as opposed to a “logical and scientific progression” [5], via “cooperative agreement” and negotiations between experts (e.g., researchers and clinicians who are members of DSM-5 or ICD-11 task forces) [6] rather than according to the validity of nosographical categories, that is, on the basis of “correspondence to external reality” [7]. Usually, once a diagnostic category has been established, if anomalies are found, the standard strategy is not to dismiss the category, but rather to aggregate it with other categories according to the principle of so-called comorbidity [8], to subdivide it into subcategories, or to add supplementary diagnostic categories. New diagnoses proliferate in each successive DSM, creating the impression of a “diagnostic sausage machine” that is somehow cranking out of control [5].

The process of diagnosis in clinical practice is compared to two people assembling a puzzle where the patient has the pieces and the interviewer has the image of the completed design [9]. The interviewer’s main goal is to discover whether a patient with a given set of signs and symptoms (i.e., the single “pieces” of the puzzle) meets the criteria (the “image of the completed design”). *Ticking boxes* is supposed to be the basic skill of psychiatrists, yet the majority of residents in psychiatry complain that they are not appropriately trained for it [10]. *Ticking boxes* seems to liken the clinician’s task to that of an accountant (counting symptoms), and “accountabilism” can be considered its epistemological hallmark. Actually, under close inspection, *ticking boxes* does not imply discovery at all (if by ‘discovery’ we mean the process of sighting the existence of something for the first time). It has no heuristic power and ends up as a mere tautology—a person with symptoms of depression is affected by a syndrome called “depression”. The main shortcomings of *ticking boxes* are “Procrustean errors” (stretching and trimming the patient’s symptomatology to fit criteria) [11] and “tunnel vision” (avoiding the assessment of phenomena not included in standardized interviews) [12].

A second method of discovery is *drafting arrows*, i.e., constructing causal diagrams: directed graphs that display linear causal relationships between variables or nodes connected by arrows to other nodes (see Figure 1). Causal diagrams are explanatory narratives, including psychodynamic or biological ones. In psychodynamic explanatory diagrams, the aim is achieving meaningfulness via the logic of motivational causation (e.g., [13]). A given symptom is the consequence of a given mind state, usually an unconscious motivation which, typically involuntarily, produces it. It is essential for a psychodynamic narrative to connect, via an arrow, a “deeper” mind state to a more “surface” or “epiphenomenal” one. Explanatory biological diagrams exploit the same logic, but in place of a psychological motivation, the cause to be linked to a given “surface” symptom is a “deep” biological anomaly.

These forms of discovery based on linear causality and linear temporality have given great strength to psychopathological knowledge, but they are not immune from limitations. Perhaps the most relevant of these limitations is that constructing causal diagrams, whether psychological or biological, transforms the sensitive and concrete presence of psychopathological phenomena, and of the person embodying them, into a world of simulacra made of unconscious motivations and hidden anatomies of functional anomalies, trading the concrete world of the senses for the abstract world of theories. This obviously implies some kind of reductionism of the complexities of human subjectivity. Although “sweeping reductionism”, a sort of “theory of Everything” according to which psychopathological phenomena are “nothing but” some “basic elements” (biological or psychological) to which they can be reduced, seems to have disappeared from the scene, another, no less problematic form of reductionism seems to be rampant. This second kind is “creeping reductionism”, which involves partial reductions that work on a patch of science, where bit by bit we obtain fragmentary explanations using disparate interlevel mechanisms including neurotransmitters, genes, neural circuits, etc. Creeping reductionism seems to tolerate a kind of pragmatic and pluralistic parallelism [14].

Is reductionism always the strategy to be pursued? Can the coexistence of multiple explanatory perspectives and their cross-level integration mitigate reductionism? How can we avoid that pluralism of causations degenerating into a disorganized list of “facts” that confuse more than they enlighten [15]? This situation created by *drafting arrows* calls to mind the image of a labyrinth, which promises that those who enter it will successfully find their way out, only to break that promise.

## 2. Emergentism: A Corrective to Accountabilism, Reductionism and Objectivism?

Last but not least, how “objective” are the narratives construed through *drafting arrows*, and how “objective” are the techniques devised to obtain reliable diagnosis through *ticking boxes*? Recognizing the value of objectivity in psychopathology and being faithful to it as much as possible is valuable, but being spellbound by the sirens of objectivity makes one lose sight of the very nature of mental phenomena, which are relational and “emergent”. Although there is no single and neat concept of emergence independent of particular explanatory contexts [16], an emergent phenomenon can be roughly defined as a phenomenon that is irreducibly relational; emergent features are not reducible to any intrinsic property of any element of a whole. Emergentism seems to be a corrective that counteracts the limitations of accountabilism, reductionism and objectivism.

Emergence occurs in the first place when an entity is observed to have properties that its parts separately do not, properties that emerge only when the parts interact on a wider whole. This is clearly ignored by accountabilism. Furthermore, traditional approaches to modeling the manifestation and evolution of psychopathological conditions often rest on the assumption that symptoms are the passive expression of a process which lies beneath it and that they can be explained by reducing them to the underlying process or to parts of it. Within an emergent framework, the genesis of psychopathological phenomena is seen from a quite different angle. For instance, in the case of schizophrenia, basic anomalous self- and world-experiences, on one side, and the patient’s resources to cope with them, on the other, face one another. Phenomenological approaches highlight the interactions between the resources of the patient, as a meaning-making agent, and his illness experience, e.g., the basic anomalous phenomena that affect him [17]. The manifestation and course of illness can be understood as emergent phenomena, the outcome of this face-to-face interaction, that is, the expression of the person’s efforts at making sense of, fighting against or adapting to the existential challenges associated with the onset of illness experiences [18]. Another example can be the way a patient experiences his illness and his awareness of being ill; a person’s awareness of illness modifies the way this person experiences the illness that affects him. Otherwise stated, and on a larger scale, the representation one has of his psychopathological condition modifies the condition itself. The illness/awareness-of-illness (or representation-of-illness) interaction implies a bi-directional dynamic and a circular causality; not only does awareness (or representation) of illness modify the illness itself, but the illness itself (especially in the case of mental illnesses) modifies one’s capacity to be aware of the illness (or one’s representation of the illness).

Emergentism is not only a corrective to accountabilism and reductionism, but also to objectivism. An emergent phenomenon is a phenomenon that does not exist in itself, but needs an observer to take place. It “emerges” thank to the interaction between something “out there” and the mind of an observer. A psychopathological phenomenon is an experienced condition whose peculiar features emerge within an interpersonal context. In different interpersonal (e.g., socio-cultural) contexts and in different contexts of care (e.g., bio-medical, psychotherapeutic or community settings), different phenomena may emerge from the same patient and they can be given different psychopathological significance. Symptom variance (which is taboo for ticking-boxes interviewing techniques) is an effect of the milieu in which a given phenomenon emerges. In the context of research, this is an opportunity to improve the characterization of a phenomenon by attaining a richer “objectivity” via the integration of multiple “subjective” views, as is the case with Consensual Qualitative Research [19]. Coproduction of scientific knowledge via the method of cowriting can be seen as an example of emergentism in psychopathological research. Cowriting can be defined as the practice in which academics and individuals with the lived experience of a disorder mutually engage in jointly writing a narrative related to the condition, sharing perspectives and meanings about the individual’s suffering [20]. The narrative produced is neither in the clinician’s mind, nor in the patient’s, but emerges from the mutual attempt to share a perspective. Diagnostic categories and pathogenetic models represent conceptual resources whose role does not consist in dispensing with the encounter with the singularity of each patient, but in orienting the clinical community in the knowledge and treatment of individual persons [21].

## 3. Linking Dots: An Alternative Emergent Way to Discovery

Should we not give up some of our claims to reliability, linear causality and objectivity and admit that discovery in psychopathology necessary produces *emergent* knowledge?

In the following, I develop an argument to support an alternative way to knowledge and discovery based on the emergent properties of psychopathological phenomena. I call this approach *linking dots*. The “dots” at issue here are basically *fragments of lived experience* provided by the patients themselves as pieces of their personal history, experienced situations, emotions, beliefs, imagination, dreams, etc., or arise in the intersubjective space between the patient and the clinician during the clinical encounter. I will refer to these fragments of lived experience or “singular phenomena” with the term “images”. The reason for choosing this word is that these “images” should be thought of as pictures to be hung on a wall, ideally posited in front of the patient and the clinician, and patiently collected before the “links” between them arise spontaneously or semi-spontaneously in the patient’s and the clinician’s minds. 

I will build on and develop the approach to images and discovery devised by art historian Aby Warburg in his atlas of images *Bilderatlas Mnemosyne* [22]. Aby Moritz Warburg described himself as “Hamburger at heart, Jew by blood, Florentine in spirit”. Born of a wealthy German Jewish family of Italian origins in Hamburg in 1866, he studied art history, history of religion and archaeology in Bonn. He also trained for two semesters in medicine in Berlin. At the end of the 19th century, he moved to Florence, where he developed an entirely original understanding of Antiquity and the Renaissance, which remained his main interest throughout his life. In the years after his mental breakdown at the end of the First World War, which shook Europe’s civilizing project at the core and sent him to Ludwig Binswanger’s psychiatric clinic Bellevue from 1919 till 1924 [23], the *Bilderatlas* became his major focus. Created in the late 1920s, its final version (October 1929) consisted of 63 wooden panels covered with black cloth, on which were pinned about 1000 pictures from books, magazines, newspapers and other daily life sources. 

This collection of images is reminiscent of a very similar project started in the same period by German Jewish philosopher and cultural and literary critic Walter Benjamin (1982–1940) [24] called *Passagen-Werk* (translated *Arcades Project*) [25], an enormous collection of quotes from heterogenous “sources” (as those used by Warburg) illustrating the city life of Paris in the mid-19th century. Benjamin’s project was inspired by an emblematic figure of the new metropolitan life: the *ragpicker*. The ragpicker, writes Benjamin, catalogues and collects everything the big city has rejected, everything it has lost, everything it has disdained, everything it has torn to pieces [25]. The analogy between the psychiatrist and the ragman leaps to the eye; both are interested in collecting and recycling the rags and scraps of human existence. It is in his attempt to reconstruct the profound transformation taking place in the Western world in the mid-19th century that Benjamin decides to adopt the principle of montage as an epistemological principle on which to base a new, anti-historical approach to the understanding and the critique of culture. The quotations collected by Benjamin in this unfinished work are neither arranged in chapters or sections, nor commented on, but simply juxtaposed with one another. It is left to the reader to link the dots.

Similar to Aby Warburg’s *Mnemosyne*, in the field of art critique, the Arcades Project is a methodological touchstone for the process of knowledge generation. The visual constellations created by Warburg in the series of panels of the *Bilderatlas* can be understood as a cultural method that uses spaces and surfaces to reveal the layers of memory and the web of relationships manifested in them. Panels invite the viewer to participate in the production of meanings, forging ever new connections between the images. It is the viewer’s acts of perception that draw relationships between singularities. This method is deemed of enormous significance in the context of today’s social and cultural crisis and transformation processes, which can no longer be comprehended using the categories of existing knowledge systems [26].

## 4. Images and Discovery

Let us briefly present the basic steps in Warburg’s method and begin to establish a parallel with linking dots in clinical practice. 

The first step consists of the establishment of an *archive* or collection of images gathered during the process of discovery (in the clinical context, the psychopathological interview) [27]. Warburg defines the units of the collection as *Pathosformeln*. *Pathosformeln* are images in which a given *pathos*—i.e., an emotional energy, an affective state—coagulates in a *formula*—i.e., a culturally transformed model [28]. Thus, *Pathoformeln* are deposits and transformers of affective/emotional drives. They give expression to the movement of life, making visible a certain emotional movement; they unite an invisible, internal state with a bodily, visible form. *Pathosformeln* create a space, a distance between the person and her internal state through a kind of objectification that shapes and begins a space for thought. 

A network of resonating images thus emerges. We may call this *Figura*. A *Figura* is a web of images reverberating with each other that is established thanks to linking dots between the individual images. Within a *Figura,* each image constitutes a nucleus for the discernment of other images that relate to it. As noted in the section on emergentism, two factors are involved in this networking process. First, the images of the archive resonate with each other. Second, the images resonate with observers’ gazes. These resonances may originate without an active, voluntary and rational contribution by the observer, or via the application of some sort of pre-established algorithm. They depend on the way the beholder *engages* with what he sees. Thus, what is revealed are not atemporal essences, but rather the outcome of the here-and-now subject–object engagement. An illuminating analogy is provided by Walter Benjamin [29]; these *Figurae* are to each image that composes them as constellations are to stars. Their prototypes are *star constellations*; just as the constellations we see in the starry night are not how stars aggregate according to the laws of nature, but according to the human eye that sees them from afar and groups them into perspicuous configurations, so too these networks are “constructed” in the eye of the beholder.

Building on such a semi-spontaneously formed network of images, the next step consists of an *active montage* of resonant images. The images are not mounted on the basis of inductive or deductive operations, but by their exemplary or paradigmatic characteristics. The juxtapositions are not even based on a diachronic temporal logic; the elements of this archive are collected without regard to the linear model of temporality. The connection between such images is *analogical*, i.e., based on similarity or ‘family resemblance’ [30]. Analogy is suspended between an emotional logic and a formal logic, and is in tension between these. 

The constellation of images highlights hidden aspects of each image that only reveal themselves in the presence of this juxtaposition. The figurative links, semi-spontaneously emerged (resonance) and actively established (montage), uncover what, in Benjamin’s words, we may call their “optical unconscious” [31]. Within this network, a *Figura* contributes to make each image more perspicuous and more meaningful. Furthermore, when individual images form a *Figura*, they also give rise to figurative *causes*. Figurative *links* within a *Figura* generate figurative *causes* since they *transform* each single image, metamorphosing it. The figurative nexus that emerges in the present retroacts on the image that comes from the past (see Figure 2). 

These are the steps of this process, but the process itself is never completed. This network of images forms an *open construction*, not a closed one such as a diagnostic category or the teleology of certain narratives. They provide a space for thought [32]. This network constitutes the space for *discovery*, for the birth of a new thought.

## 5. Conclusions

*Atlas Mnemosyne*, Warburg’s masterpiece, is a collection of images built to shed new light on Florentine Renaissance art. It contains several groups of images pasted on large panels, designed to find and show analogies between the various images grouped in this way. These *Figurae* are visual coinages that are established thanks to the pathic (rather than conceptual) resonance and interference between images. I argued that this practice could be taken as an exemplary method of discovery to be applied in the field of psychopathology, where the phenomena at issue are *pathic* and *relational* in nature. I suggest that Warburg’s method is an illustration of linking dots, an alternative to (or an integration of) other methods such as *ticking boxes* and *drafting arrows*. 

In these concluding remarks, I would like to summarize the epistemological principles of this method of discovery. Linking dots proposes to shift attention from symptoms conceived as objective, conceptually described “facts out there” to non-schematized affective qualities arising in a relational space condensed in images (*Pathosformeln*).

It also recommends to move away from computational and narrative connections between phenomena, and towards analogical figurative links between them within a network of analogies constituting a space for discovery.

Last, but not least, it advises to focus on, rather than linear causality, circular relationships and the emergence of figurative causes generated via the interaction between the parts of a whole, and between these parts and the observer.

Whereas in the logic of *ticking boxes*, the particular (the symptom) represents a regularity by identifying to which general diagnostic category a patient’s individual case can be traced back, in the logic of linking dots, the particular (a fragment of lived experience, a single image) does not have its own intrinsic meaning (e.g., as a diagnostic index) that refers to the general, but instead represents an exception, a singularity that arouses perplexity and disrupts the previously acquired representation of the patient’s condition, to the point of threatening the stability of the diagnostic categories. The objective of gathering images in the logic of linking dots is not to *decipher* a hieroglyphic, but to *produce* a thought-provoking hieroglyphic [33].

Obviously, this method is not meant to propose a new classification of psychopathological phenomena. Rather, one of its aims is to challenge current diagnostic categories. Furthermore, this method of discovery may be more suitable for clinical care than for research purposes. Nonetheless, it may open new paths of research by describing unresearched or under-researched psychopathological phenomena.

## Figures and Tables

**Figure 1 brainsci-13-00013-f001:**
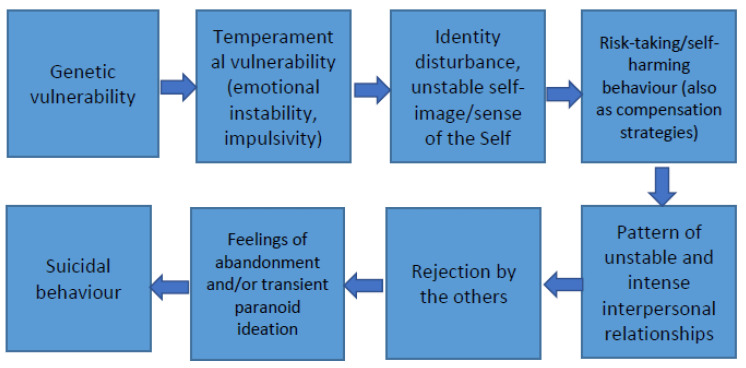
*Drafting arrows.* A biological–psychological interface model for borderline pd.

**Figure 2 brainsci-13-00013-f002:**
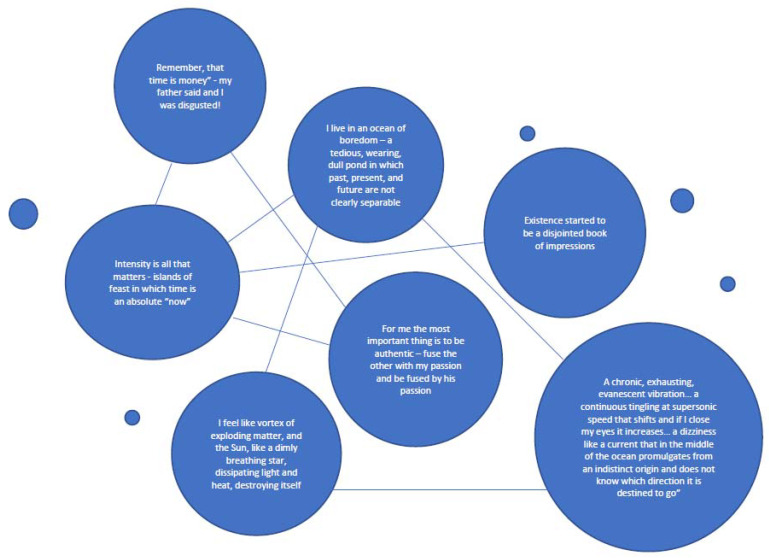
Linking dots. Images from a dissipating style of existence.

**Table 1 brainsci-13-00013-t001:** *Ticking boxes*. Diagnostic criteria for borderline personality disorder.

Chronic Feelings of Emptiness
Emotional instability in reaction to day-to-day events (e.g., intense episodic sadness, irritability, or anxiety usually lasting a few hours and only rarely more than a few days)
Frantic efforts to avoid real or imagined abandonment
Identity disturbance with markedly or persistently unstable self-image or sense of self
Impulsive behavior in at least two areas that are potentially self-damaging (e.g., spending, sex, substance abuse, reckless driving, binge eating)
Inappropriate, intense anger or difficulty controlling anger (e.g., frequent displays of temper, constant anger, recurrent physical fights)
Pattern of unstable and intense interpersonal relationships characterized by extremes between idealization and devaluation (also known as “splitting”)
Recurrent suicidal behavior, gestures, or threats, or self-harming behavior
Transient, stress-related paranoid ideation or severe dissociative symptoms.

## Data Availability

Not applicable.

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
