# Peer review of "The Power of Images and the Logics of Discovery in Psychiatric Care"

_brainsci, 2022, doi:10.3390/brainsci13010013_

Round 1

Reviewer 1 Report

Review for Brain Sciences, Manuscript ID brainsci-2088798-peer-review-v1: Logics of Discovery.

Dr. Stanghellini eloquently highlights the limitations of our reductionist approaches to psychiatric care, offering an alternative, “emergent” perspective and approach.  The paper is a delightful read, nearly poetic with its images and analogies, despite some English-language errors that do not affect the overall content and its readability.

I expect that this paper will be warmly received by the growing community of psychiatrists, psychologists, and philosophers who recognize the difficulties with applying reductionist methods borrowed from the natural sciences in describing the inner, subjective world of human experience, the primary target of psychiatric interventions.  The paper is well aligned with modern thinking in terms of patient-centered care and co-creation of patient care; and it may inspire research that better characterizes human experience, as advocated by Alvan Feinstein.

Minor comments:

Details are needed for Supplementary Materials, Funding, Acknowledgments, and Conflicts of Interest.

Author Response

I am really very grateful to Reviewer 1's comments and encouraged by his/her appreciation to continue my research in this field. I will send my paper to a native English speaking expert to improve my English. I will also add details for Supplementary Materials, Funding, Acknowledgments, and Conflicts of Interest as suggested.

Reviewer 2 Report

Review for Manuscript ID: brainsci-2088798 entitled " Logics of Discovery”

The manuscript is not well written and there is minor points need to be addressed as follows: 

1.    The title must be amended as it does not represent the content.

2.    The abstract is not clear and does not show a final conclusion. 

3.    Pronouns are heavily used. These have to be amended and the sentence is better to be in the passive tense.

4.    I think it is better to have Ticking boxes and drafting arrows as an example figure.

5.    Line 87, what is “eg”

6.    I would like to have a figure for the author’s claim about linking dots. 

BW, 

Author Response

  1. The title must be amended as it does not represent the content. REVIEWER 2 IS ABSOLUTELY RIGHT ABOIT THAT. INDEED, I WAS ASKED TO DELETE THE SUBTITLE FROM MY FIRST SUBMISSION. THE NEW TITLE IS “THE POWER OF IMAGES AND THE LOGICS OF DISCOVERY IN PSYCHIATRIC CARE”
  2. The abstract is not clear and does not show a final conclusion. I THANK REVIEWER 2 FOR THIS COMMENT. I HAVE ADDED, AT THE BEGINNING OF THE ABSTRACT, THE FOLLOWING SENTENCE: This paper, aligned with contemporary thinking in terms of patient-centered care and co-creation of patient care, highlights the limitations of the reductionist approaches to psychiatry, offering an alternative, “emergent” perspective and approach.
  3. Pronouns are heavily used. These have to be amended and the sentence is better to be in the passive tense. I ASKED AN EXPERT ENGLISH NATIVE SPEAKER TO CORRECT MY PAPER.
  4. I think it is better to have Ticking boxes and drafting arrows as an example figure. I ADDED THREE TABLES, ONE FOR TICKING BOXES, ONE FOR DRAFTING ARROWS AND ONE FOR LINKING DOTS
  5. Line 87, what is “eg” I HAVE CORRECTED IT NOW IT IS e.g.
  6. I would like to have a figure for the author’s claim about linking dots. SEE RESPONSE TO POINT 4

Reviewer 3 Report

This is a most outstanding masterpiece of research in psychopathology. It delivers an overview on the dominant perspectives and narratives in constructing psychiatric diagnosis, namely "ticking boxes" (meeting predefined diagnostic criteria) and "drafting arrows" (defining linear causal relationships).

The author then argues for alternative approach, "linking dots", based on emergentism concept. The diagnostic construct here emerges from constellations of resonant images from an atlas by A. Warburg.

This manuscript is a brilliant chef-d'oevre of the intellectual synergy between arts, humanities, and psychopathology.

For any single word or sentence in and out of the text would damage its holisitic, symphonic structure.

Perhaps it may benefit from insertion of few images from the Bilderatlas Mnemosyne as figures.

It was a priviledge for me to review it.

Author Response

I am delighted to read this review and extremely grateful for the Reviewer's appreciation of my work. I do hope we will be in touch in the nearest future.